# Continually Updating Neural Causal Models

**Florian Peter Busch**[†1,2]    **Jonas Seng**[1]    **Moritz Willig**[1]    **Matej Zečević**[1]

[1] Computer Science Department, AIML, TU Darmstadt, Germany
[2] Hessian Center for AI (hessian.AI)
[†]correspondence: `florian_peter.busch@tu-darmstadt.de`

## Abstract

A common assumption in causal modelling is that the relations between variables are fixed mechanisms. But in reality, these mechanisms often change over time and new data might not fit the original model as well. But is it reasonable to regularly train new models or can we update a single model continually instead? We propose utilizing the field of continual learning to help keep causal models updated over time.

## Introduction

Causal models (Pearl 2009; Peters, Janzing, and Schölkopf 2017) can be useful in a variety of applications (Koch, Eisinger, and Gebharter 2017; Carriger and Barron 2020; Fenton et al. 2020). Various algorithms exist for causal discovery, i. e. learning the causal graph from data (Glymour, Zhang, and Spirtes 2019). With the causal graph known, a causal model can be trained (parameter learning), enabling a causal perspective on the respective problem. Due to a lack of knowledge about unobserved variables, the relations between effect(s) and cause(s) are usually not deterministic, instead relying on probabilities for making predictions.

Continual learning is another research direction which in recent years has attracted the attention of an increasing number of people (based on the # of publications, see the discussion by Mundt et al. (2022)). While we are not aware of any universally accepted definition, we would describe continual learning roughly as updating a model over time, given new information and without losing the *knowledge* encoded in the original model (Mundt et al. 2022; Chen and Liu 2018). The general idea behind this paper is that continual learning can help causal models stay up-to-date over time.

Imagine there is a causal model trained on some data and we assume that this is the best possible model that can be obtained using known techniques and the provided data. What if something in the problem changes or we get new data describing only some parts of the true underlying model? For example, consider a company trying out different products to sell. Every day, they put a different type of product (A) up for sale and at the end of the day, they obtain information about the amount of money earned (B). In this example, the company is still trying to find out which products are selling well and which do not, so it is safe to assume that the amount of money earned here does not causally influence the type of product sold. On the other hand, which products are put up for sale certainly determines the profit at the end of the day. Therefore, we have the causal graph A → B and, given enough data, it can be calculated how much profit is to be expected depending on the type of product sold. Most likely, in a real-world scenario, there would be more variables included which have an influence on the model, for example the number of guests entering the store or the day of the year. We exclude most of these in our example but it is reasonable to assume that some of those would be included and known, while other variables which influence the model are not included, either because the actual values are unknown (e. g. wealth of the customers that day), they might not have been thought of, or it was impossible to include them for other reasons. Now, imagine one such variable suddenly becomes available. For example, the company could sell its wares in several places and one day the data scientists get access to new data including the location of the store (C) which now is another cause of the amount of money earned (new causal graph: A → B ← C). How can the existing model be updated to account for the newly acquired data without having to retrain the entire model?

But even if the general structure of the problem (causal graph) remains unchanged, we could benefit from continual learning. Staying with the previous example, imagine that the popularity of a product type changes. We would assume that much of the relations between the causes A and C and the effect B remains the same but certainly some relations including that specific product type change.[1] Is it possible to retain the knowledge which still holds and at the same time update the model to represent the new situation?

In this paper, we focus on Neural Causal Models (NCMs) introduced by Xia et al. (2021) as a parameterized form of general Structural Causal Models. Here, continual learning provides tools which can help keep NCMs up-to-date.

First, we give a short introduction on causal models, followed by a more detailed explanation of the considered problem and an overview of general strategies to solve it.

---

[1]Arguably, this is not a change of relations but a change of unknown variables not included in the model (lack of knowledge). Even so, it can be useful to think of it as a change of relationships.

Before concluding, we sketch out ideas on how existing continual learning methods can help tackle the problems mentioned above.

## Causal Models

Following the definition by Peters, Janzing, and Schölkopf, a *Structural causal model* (SCM) is given by $\mathfrak{C} := (\mathbf{S}, P_{\mathbf{N}})$ where $P_{\mathbf{N}}$ is a joint distribution over all noise (exogenous) variables and $\mathbf{S}$ is defined as a set of $d$ structural assignments

$$X_i := f_i(\mathbf{PA}(X_i), N_i), \quad \text{where } i = 1, \ldots, d, \quad (1)$$

where $\mathbf{PA}(X_i)$ are the parents of $X_i$, i.e. those variables in the causal graph which have an edge ending in $X_i$ (the variables which *cause* $X_i$), $N_i \in \mathbf{N}$ is a noise variable, and $d$ represents the number of variables.

Neural causal models (NCMs) are a type of SCMs where the functions $f_i$ are given by feedforward neural networks (NNs) (Xia et al. 2021). Please refer to the original paper for a full definition of NCMs.

## Problem

The problem addressed in this position paper is concerned with updating causal models given new information (data). Here, it can be distinguished between two general problems:

**P1** Parts of the probability distributions of the model change (the causal graph remains unchanged).

**P2** The structure of the causal graph changes (i.e. variables are added or removed or other cause-effect relations in the causal graph change).

In mathematical terms, if $f_i^{\text{old}}$ is the structural assignment for variable $X_i \in \mathbf{X}^{\text{old}}$ with parents $\mathbf{PA}^{\text{old}}(X_i)$, and noise distribution $P_{\mathbf{N}}^{\text{old}}$ of the original model and $f_i^{\text{new}}$, $X_i \in \mathbf{X}^{\text{new}}$, $\mathbf{PA}^{\text{new}}(X_i)$, and $P_{\mathbf{N}}^{\text{new}}$ are the respective variables and functions in the desired new model, then **P1** describes the problem of changing $f_i^{\text{old}}$ to $f_i^{\text{new}}$ given that $\mathbf{X}^{\text{old}} = \mathbf{X}^{\text{new}}$ and $\mathbf{PA}^{\text{old}}(X_i) = \mathbf{PA}^{\text{new}}(X_i)$. Both of these requirements do not have to hold for **P2**.

Since **P2** usually includes **P1**, **P1** can be seen as a sub-task of **P2** and the overall easier problem.

But why is this an important problem? Is it not easy to simply add the new data to the original data or replace some parts of the original data and retrain the model? Depending on the application, this might be a valid possibility and, in that case, continual learning is not needed. However, just retraining the model has several possible downsides:

**A) Time and efficiency in general.** Training a new causal model can take a lot of time and resources. Especially if the data changes regularly, it could be unreasonable to train a model from scratch every time.

**B) Original data is unavailable.** Privacy aspects, storage constraints, and other reasons could make it impossible to keep data stored for a prolonged time period. If the new data is not sufficiently large and complete, a retrained new model could end up performing significantly worse than the original model, while an updated model could benefit from both the information of the original model (indirectly the original data) and the newly acquired data.

**C) New data is incomplete.** The new data might not be complete and only contain some features, like a new variable or only the features for one cause-effect relationship which presumably changed. Here, retraining could simply be impossible.

## Proposed Solution Strategy

First, it is worth mentioning that simply using a model such as an NCM inherently (to a degree) opens the causal model up to continual learning. Since the "mechanisms" (functions determining a variable based on the parent variables) are usually assumed to be independent of each other, they can also be updated separately. In other words, if it is known that only a certain subset of mechanisms changed, those can be updated while requiring data only for the features (variables) relevant for these mechanisms (the respective child and parent variables).

This can be very useful but is not an exciting new revelation so let us get back to the two aforementioned problems and discuss solution strategies. For the second, more complex problem, these strategies are less specific but might serve as first steps towards tackling that problem.

In the following, some general solution strategies are introduced. Examples of specific continual learning methods and how they could be useful are discussed afterwards.

### Problem 1: Change of Probabilities

**Retraining.** Training a new model is a valid strategy in general but this approach also has various problems (refer to the previous parts of this paper).

**Continue Training.** One can continue training as before but using the new data. If what is trained on now consists of data representative for the full data distribution, this should work out well. However, if the new data is very specific and does not capture the full range of the data, *catastrophic forgetting* (Robins 1995; Kirkpatrick et al. 2017) could become a problem, where predictions for data points not represented by the new data are incorrect, although they were correct for the original model.

**Continual Learning.** Continual learning can help a lot, depending on the specific problem formulation. Assume that we have discrete variables[2] and therefore, given a specific model, there is only a finite amount of probabilities this variable can obtain (one for each parent configuration). If an NCM is used and the new data only covers some of these parent configurations, continual learning methods can be used to avoid (or at least reduce) *catastrophic forgetting* of the other parent configurations.

### Problem 2: Change of Structure

Continual learning could also be a helpful tool towards updating the structure of a causal model. If a variable is added as a new parent to another variable, the existing NN could

---

[2]The idea could also work for continuous variables but it requires a more sophisticated approach. For this position paper, we consider discrete variables as the simpler version of this problem.

be extended by additional neurons to increase the expressivity of the NN and maybe even keep some useful connections and neurons within the NN which are still helpful (but they should not be fixed in case the relationship changed significantly). In general, adapting the NN architecture instead of retraining might enable the new model to utilize what the original model has learned while at the same time being trained with new information.

## Continual Learning Methods to Help NCMs

There are many ways in which continual learning methods could help update NCMs when retraining is impossible or too expensive. A rather straightforward approach for keeping some information encoded by the original model when training on a new dataset is given by rehearsal and pseudorehearsal (Robins 1995). Here, in addition to the new data, the model is trained on specific instances either from the original dataset (rehearsal) or artificially created ones (pseudorehearsal). The idea is that those instances are especially representative for the original model behavior. But there are also methods for which no training on anything but the new dataset is necessary. Sharing the goal of keeping previously learned model behavior, elastic weight consolidation (Kirkpatrick et al. 2017) can be used to slow down learning for neurons responsible for specific input-output relationships to reduce forgetting. For improving an existing model with new information, few-shot learning methods (Wang et al. 2020) might be of use when the amount of data to update the model is small. Maybe the same goal can even be achieved by zero-shot learning (Xian et al. 2018) without needing a single data instance but how one would go about this exactly requires further thought. Perhaps training a new model is possible and even desired but the original data is not available anymore. In that case, knowledge distillation (Gou et al. 2021) could be used to distill input-output relationships (maybe even excluding those that are deemed outdated). One could even go one step further and not only use knowledge distillation to update the probability distributions of the model (**P1**) but also create the new model in such a way that it accounts for a changed causal structure while keeping useful information from the previous model (**P2**). But there are also other continual learning methods which could be useful when the causal graph changes.

Looking at a relatively simple aspect of that problem, transfer learning (Pan and Yang 2010) might be useful when a new variable is added to a NN with the same inputs (causes) as another NN because the way these causes impact the effect variables (outputs) might be somewhat similar. In the event of adding an additional effect instead of another cause, if a relationship between some variables is presumably complex but has two or more effects (i.e. child variables), maybe those could benefit from sharing parts of the NN architecture at the beginning, while having different child variables represented by different output (task) specific layers at the end of the NN (Li and Hoiem 2017). In other words, adding a new effect would keep the previous layers unchanged and only require one to add and train the new output specific layers. More generally, if a NN in an NCM does not have enough capacity to cover the old and new data

or if new variables are added to the model, expanding the NN architecture (Rusu et al. 2016; Yoon et al. 2018) can be a promising alternative to training an entirely new model. Expanding instead of retraining allows this approach to also benefit from what the previous model had learned.

Let us consider an high-level example inspired by the introduction to illustrate how updating NCMs could look like.[3]

1. A dataset $\mathbf{D}_{A,B,C}$ contains a lot of data points over the features A (type of product) and B (money earned). An NCM $\mathbf{M}_0$ is trained given the causal graph A $\rightarrow$ B.

2. Due to some reason (e.g. privacy), $\mathbf{D}$ is deleted.

3. A new, small dataset $\mathbf{D}_{A,B,C}$ becomes available, also containing information about C (location of store).

4. By adding neurons to the structural equation, the information in the previously learned NCM $\mathbf{M}_0$ can be utilized to train $\mathbf{M}_1$ (for example using progressive neural networks (Rusu et al. 2016)).

5. Store owners report that a specific product $a_i \in$ A lost much of its popularity (maybe some kind of scandal) and a new dataset is created $\mathbf{D}_{a_i,B,C}$ only for product $a_i$.

6. With the help of continual learning, the NCM $\mathbf{M}_1$ can be updated in such a way that mostly the neurons responsible for the input A $= a_i$ change (for example using elastic weight consolidation (Kirkpatrick et al. 2017)), while other neurons stay relatively unchanged.

## Conclusion and Outlook

One might ask why causal model would even need to be changed over time. Do true cause-effect relationships ever really change? In order to be useful, causal models usually follow certain assumptions and are only abstractions. It is unreasonable to try to go into the "most detailed cause" on an atomic level. Therefore, the continual aspects come into play when either the available information changes (the "true laws of our universe" stay the same, but those are not modeled explicitly, only a certain abstraction is) or a part of the problem itself changes. Again, in this second case, no atomic cause-effect relations change but our chosen abstraction does not fit the current scenario anymore. For example, this applies if the i.i.d. assumption is violated.

Continual learning and causality (NCMs in particular) have several goals in common, including but not limited to model adaptation given new information, invariance of unchanged knowledge, and efficient use of data. We postulate that causal models can benefit from continual learning methods which are designed to update new or changed parts of a model while keeping other parts functional.

One can also think of further areas in which a continual perspective on causality could help. For example, one might even try to create some kind of "meta model" which operates on top of a causal model but, given the previous changes in that causal model, is tasked to predict how the causal model is expected to change in the future.

---

[3]Of course, this example containing only 2 to 3 variables is very simple but it should make clear what kinds of problems could be solved using continual learning and how one could go about it.

**Acknowledgments** The authors thank the anonymous reviewers of the Bridge program for their valuable feedback. Furthermore, the authors acknowledge the support of the German Science Foundation (DFG) project "Causality, Argumentation, and Machine Learning" (CAML2, KE 1686/3-2) of the SPP 1999 "Robust Argumentation Machines" (RATIO). This work was supported by the Federal Ministry of Education and Research (BMBF; project "PlexPlain", FKZ 01IS19081). It benefited from the Hessian research priority programme LOEWE within the project WhiteBox, the HMWK cluster project "The Third Wave of AI" (3AI) & the National High-Performance Computing project for Computational Engineering Sciences (NHR4CES).

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
