# OpenReview forum: "Continually Updating Neural Causal Models"
_AAAI.org/2023/Bridge/CCBridge — AAAI23 Bridge Continual Causality_

### Official Review · Reviewer_VwG1 · 2022-11-29

**Rating:** 7
**Confidence:** 4

**Review:**

This position paper described two problems for continual learning of causal models and proposed solution strategies for these problems. The suggested solutions are interesting and can potentially utilize existing continual learning methods for these problems. It would also be interesting to see how some continual learning algorithms (such as the replay-based algorithms) can be adapted to causal models.

---

### Official Review · Reviewer_inNQ · 2022-11-30
**Interesting high level setting**

**Rating:** 7
**Confidence:** 5

**Review:**

Summary:

This proposal is about learning causal mechanisms and models as we collect new data where new variables are measured. In this problem, while the true underlying structural causal model (SCM) remains invariant, the revealed parts of the model might sequentially change, meaning that our estimates of different conditional distributions will change. The research of interest in this proposal is to use ideas from continual learning to update a class of SCMs called neural SCMs.

Significance: The motivating problem and the high-level setup are compelling, and could be impactful for many domains. Further, the idea that variables that are measured change as we get new data is significant for learning causal models and mechanisms.

Weaknesses: While I found the running example with three variables helpful, I think that the main challenge for the authors is to formalize their problem more technically and clearly. Currently, the problem is framed at too high a level, about accounting for new variables in causal models. But it's not clear what this high level problem means in terms of concrete causal estimands. E..g, how do new variables affect the learning of causal conditionals, how do they affect causal structure learning, etc.

My recommendation to the authors is to focus first on formulating at least one concrete mathematical question and then adapting techniques from continual learning.

---

### Official Review · Reviewer_PW7z · 2022-11-30
**Suggestions**

**Rating:** 6
**Confidence:** 4

**Review:**

Given the length requirements, I think this paper is a bit challenging to read coming from a continual learning background. I think briefly introducing causal models, and specifically NCMs would be useful. Then motivate why they would benefit from continual learning. While there are a plethora of continual learning paradigms, many aren't appropriate unless there are specific restrictions on either memory or more rarely compute, and many make assumptions about whether task labels are available and the sequence order. I think the example at the beginning is good in terms of why have a continual learning model where the store changes its inventory daily, but the reasoning behind why continual learning is needed is a bit lacking. Again, I'm ignorant about causal models, but perhaps listing the difficulties and challenges they face and then how continual learning could help overcome those challenges would be helpful. Depending on the complexity of training causal models, that could motivate using continual learning, but if it is relatively cheap to train them then just growing a database and periodically re-training would suffice. I think some argument for why that approach is lacking for NCMs and extremely sub-optimal for use cases would be helpful.

I think some care and thought into the continual learning methods that are most appropriate and least appropriate for NCMs would be helpful, rather than just listing techniques.

---

### Decision · Program_Chairs · 2022-12-05

**Decision:**

Accept

**Comment:**

Accept - Poster

The paper discusses whether continual learning provides benefits to causal machine learning compared to retraining a model from scratch. The paper is of interest to the bridge and highlights some initial first steps in combining approaches from the two fields. We encourage the authors to include more discussion about how the causal aspects of the paper relate directly to approaches from the continual learning community.